# New Zealand Youth Rugby Sevens: A Comparative Match Demands Study

**DOI:** 10.3390/jfmk8020041

**Published:** 2023-03-31

**Authors:** Koen Wintershoven, Christopher Martyn Beaven, Nicholas David Gill, Daniel Travis McMaster

**Affiliations:** 1Te Huataki Waiora School of Health, University of Waikato, Adams Centre for High Performance, Mount Maunganui, Tauranga 3116, New Zealand; 2All Blacks, New Zealand Rugby Union, Wellington 6011, New Zealand; 3All Blacks & Black Ferns Sevens, New Zealand Rugby Union, Wellington 6011, New Zealand

**Keywords:** match analysis, movement demands, U15/U19, small-sided games, rugby union, kinematics

## Abstract

Rugby sevens has established itself on the world stage since its inclusion in the 2016 Olympics. Participation among New Zealand (NZ) youth has surged. Sevens games have specific high demands, but little is known about these competitive demands in regards to youth. Two NZ male youth squads (U15, n = 13; U19, n = 14) were monitored during a national sevens tournament. Microsensor technology captured heart rate (HR) and kinematic performance. The rating of perceived exertion (RPE) was collected for U15 matches only. U19 and U15 players ran 108 ± 11 and 116 ± 13 m·min^−1^ at an average speed (V_AVG_) of 6.5 ± 0.6 and 6.9 ± 0.8 km·h^−1^. Peak speeds (V_PEAK_) reached 33.7 km·h^−1^, and high-intensity running distance (HIRD) averaged 252 ± 102 m. U15 (44.3 ± 9.2 game^−1^) and U19 (39.4 ± 6.1 game^−1^) showed different sprint rates. U15 covered more moderate-velocity distance (20–80% V_MAX_) and less low-velocity distance (<20% V_MAX_). RPE was 13 ± 1 (U15). An average HR of 90.0 ± 3.9% HR_MAX_ was recorded. Upwards of 57% of game time was played at >95% HR_MAX_. Youth sevens competition is specifically demanding. U15 can experience greater loads than older peers in rugby. Coaches can use this information to optimize players’ physical development.

## 1. Introduction

The game of rugby sevens, commonly referred to as sevens, is played between two teams of seven players across two halves of 7 min, with a 2 min break interval, on a full-sized rugby pitch [1,2]. Although the basic rules of play are the same as in rugby union (RU), the competition differs from other codes in its format; every team will face several opponents per day in multi-day tournaments (typically 2–3 days) [1,2,3,4]. Sevens has surged in popularity since its introduction to the 2016 Olympics [5], which is exemplified by the 40 teams that were hosted at the 2022 RWC sevens [6,7]. This rising interest in sevens is also evident among youth; eleven nations from World Rugby’s six regional associations participated in the 2018 Youth Olympics [8], and while student sports participation in New Zealand (NZ) has dropped from 56% to 48% between 2000 and 2020, engagement in sevens grew by 185% for secondary school-age youth between 2012 and 2020 [9,10].

Sevens competition is characterized by high relative distance (RD) and intermittent high-intensity running (HIR), in support of specific defensive and offensive skills, which encompass impact-related activity [1,11] The impact on players has been stated to be particularly high, warranting differential investigation into sevens [11,12,13,14,15,16]. These specific and high match demands have to be recurrently and consistently reproduced, due to the tournament format [16]. Therefore, a sevens cohort-specific approach to preparation, based on the game demands, is necessary [11,17]. The evidence on sevens game characteristics is biased towards elite seniors, and little is known about youth sevens competition [16,17,18]. Two studies have included time-motion (TM) analysis on youth, reporting total distance (TD) per game of 1213 m by junior elites in an Australian national tournament, and 1133 m to1185 m in a South African (SA) provincial tournament [19,20]. Players appear to cover most of that distance walking (429 m) and sprint six times per game, while reaching maximal speeds (V_MAX_) of 31 km·h^−1^. No heart rate (HR) or rating of perceived exertion (RPE) were documented. Differences between age groups, playing level, and geographic location seem evident [19,20]

No studies have investigated the NZ school-age sevens population. Specific match characteristics exist for rugby union youth, differentiating school age groups and senior players [21]. It is unclear how the game demands evolve during the sevens developmental pathway. Junior-level sevens should be targeted in investigations to better represent the playing population and adequately prepare these athletes [2,17,22,23]. Accordingly, this study aimed to measure, describe, and compare the differences in movement, physiological, and physical demands between two secondary school-age youth cohorts competing in the national sevens competition. This data will inform practitioners about the age-related match demands of youth sevens players and can help optimize their specific development.

## 2. Materials and Methods

### 2.1. Experimental Approach

A cross-sectional observational study was conducted on two NZ youth rugby cohorts. Inclusion criteria for purposive sampling were the availability of two representative secondary school academy cohorts, competing nationally in different age categories. Perceived intensity, heart rate (HR), and TM data were collected using wearable VX SPORT™ microsensor technology (VXWR5Lb; 10 Hz GPS, Glonass, QZSS, SBAS, Galileo and BeiDou compatible; tri-axial 100 Hz accelerometer; 18 Hz magnetometer; 18 Hz gyroscope), coupled Suunto HR monitors, and the Borg scale (6–20) [24] on two single days during a national rugby sevens tournament. Ethics approval was obtained from the University of Waikato Ethics Committee (HREC(Health)2019#16).

### 2.2. Subjects

Two male cohorts competing in their respective age categories, under 15 (U15: n = 13; age 14.9 ± 0.3 [range 14–15]; height 175.0 ± 7.0 cm; mass 75.0 ± 8.0 kg) and under 19 (U19: n = 14; age 16.9 ± 1.2 [range 15–18]; height 179.0 ± 6.3 cm; mass 88.0 ± 9.8 kg), were involved in this study. Selection and procedures were coach-led, with no researcher interference. Subjects were informed through written and verbal means about the purposes, methods, and procedures of the study prior to their participation. Written consent was obtained from the subjects and their legal guardians.

### 2.3. Data Collection Procedures

Subjects were initiated in the use of the microsensor units and the Borg scale during the late 2020 rugby season, within a month of the tournament [24]. The units were worn between the scapulae in purpose-built vests under the playing jerseys. Similar technology, including VX SPORT™, has been applied to team sports, the football codes in particular, and to rugby youth and sevens research, to investigate kinematic demands and quantify performance [20,21,25,26,27,28,29,30,31]. A review of the literature shows that the degree of validity, reliability, and accuracy is dependent on the specific outcome measure, but a minimum sampling frequency of 10 Hz GPS is generally accepted to be adequate for measuring most rugby-related movement demands [32,33,34,35,36,37,38,39,40]. A mean value of 77 ± 6% for positioning quality during match-play has been demonstrated for similar equipment [41]. 

Players’ top speed (V_TOP_) (40 m straight-line maximal sprint) and maximal heart rate (HR_MAX_) (a rugby-specific 1200 m shuttle run, i.e., “bronco”) were established using these microsensor units following gear initiation [42,43]. Body weight and height were collected by the coaches at the start of the tournament. Microsensor units were activated before their allocation to players, pre-warm-up—approximately 45 min before games—to secure a signal (4–12 satellite connections). Normal team procedures were then followed. 

Data was captured for four U19 and three U15 games in summer conditions (18–24 °C; 60–86% humidity, uncovered dry natural grass pitches). Games were approximately 1.5–2.5 h apart. Playing halves lasted 7 min, and retrospective analysis revealed that half-time breaks ran up to 5 min. Rolling substitution was applied. RPE was collected verbally from individual players within 15 to 30 min post-game, during group debriefing. No RPE was collected for U19 due to inconsistent player availability.

### 2.4. Data Handling Procedures

Microsensor data was downloaded using the manufacture’s software (VX View version 5.4.3.53). Individual players’ kinematic and physiological data were visually assessed and manually trimmed to solely include actual playing time. Fourteen corrupted HR and four kinematic data files (technical malfunction or misuse) were identified via numerical and graphical assessment, and subsequently removed. Individual data files were coded for playing minutes by rounding up or down to the nearest full minute. There were a limited number of complete 7-min playing bouts (30/53 U15; 46/71 U19) due to coaching decisions, continuous substitutions, and long half times that resulted in modified playing time. We decided to exclude twenty-two bouts that were shorter than a quarter of a game (<four minutes) from our analysis because very short bouts are not representative of typical game demands and are known to overestimate movement variables [20,44].

Up to 54 (U19) and 44 (U15) data files were included for analysis of respective outcome measures across all games. Variables and metrics commonly used in rugby were selected as available, and labels were adopted from the software package; specific variables and categories were consequently reported as relative to individual players’ maximal values [17,18,29,45]. V_TOP_ and HR_MAX_ were based on the highest absolute recordings during baseline testing or matches, whereas maximal speed (V_MAX_) refers to the highest speed performed during games. Where players’ HR_MAX_ was unavailable, a theoretical age-specific HR_MAX_ was calculated using (208–0.7x × age) [46]. HIR category labels were adopted from the available software options and set to converge with benchmarks put forward in systematic research [17]: high-intensity running distance (HIRD > 18 km·h^−1^), high-speed running distance (HSRD > 20 km·h^−1^), and very high-speed running distance (VHSRD > 25 km·h^−1^). High-intensity acceleration (HIacc) and deceleration (HIdec) thresholds were kept to the manufacturers setting of 3 m·s^−2^, based on prior sevens research [20]. Relative speed zones (V_Z_) provide a more appropriate perspective on youth and lower-level play [47]. The V_Z_ were defined as <20% (z1), 20–50% (z2), 51–80% (z3), 81–95% (z4), 96–100% (z5) V_TOP_ [25,48,49]. Relative HR zones (HR_Z_) were adapted from Higham et al. (2016), based on Edwards (1993), to be <59% (z1), 60–69% (z2), 70–79% (z3), 80–89% (z4), and ≥90% (z5) HR_MAX_ [45,50]. Further explanations regarding metricsterminology and definition are available in the VX Sport metrics glossary [51].

### 2.5. Statistical Analysis

Established data points were exported to an Excel worksheet (Excel 2016, Microsoft Office) and statistically analyzed using the additional Data Analysis ToolPak. All absolute data was normalized to per minute, i.e., relative, and per full game total values, i.e., 14 min, for meaningful comparison. Descriptive statistics are reported as mean ± standard deviation (SD). Differences within and between groups were calculated using independent sample T-test and single factor ANOVA. Heterogeneity was checked through F-tests. Statistical significance was set at *ƿ* ≤ 0.05. For multiple pairwise comparisons, Bonferroni correction was applied [52]. Standardized mean changes were also calculated as Cohen’s *d_s_* effect size (ES)*,* using the pooled SD, to indicate trivial (<0.2), small (0.2–0.59), moderate (0.6–1.19), large (1.2–1.99), very large (2.0–3.99), and extremely large (≥4.0) magnitudes of difference [53,54]. 

## 3. Results 

A pooled set of 98 data files (U15 + U19), i.e., playing bouts (playing time across all active players), were identified with an average duration of 5.8 ± 1.8 min. Twenty-two bouts did not meet inclusion criteria for duration and were excluded from further analysis. Mean game movement demands for cohorts are reported in Table 1. Thirty-five post-game RPE values were collected across three U15 games for players’ respective playing bouts. As the term “impact” is a function of its definition and threshold values, and recognizing the importance of considering the influence of physical contact on perceived exertion, we want to provide the exact context for the impact rate discussed in this study [55,56]. We furthermore report “body impacts” as the range observed for all actual playing bouts recorded, to be U15 [0–4] and U19 [0–7], on average. Most of the distance was covered at low relative speed (<50% V_TOP_) by both cohorts, as reported in Table 2. Table 3 shows that most of the full game time was performed at very high HR, relative to the players’ maxima. U19 placed 9/31 and U15 placed 5/29 in the final tournament ranking. 

## 4. Discussion

To the best of our knowledge, this is the first study characterizing the sevens match demands in the NZ educational pathway. NZ school sevens competitions were found to contain HIR and cardiovascular demands, as compared to average sevens values and representative RU age groups. These findings do not correlate align seamlessly with perceived effort (U15). Differences were found between U15 and U19 for TD, RD, V_AVG_, sprint rate, and ActivityLoad 3D rate (AL_3D_). The distance covered in V_Z_ differs between age groups, whereas times spent in HR_Z_ is similar.

Within this investigation, TD was as high as that of international-level male sevens players (1574.4 ± 267.4 m) [57]. In comparison, these youth sevens cohorts covered up to 39% of the TD of U16 in RU matches, in 18% of the game time [21]. U15 covered significantly more TD per game than U19 (~110 m). The results also demonstrate a larger inter and/or intra-player variability at younger ages. Parallel to this, U15 obtained a higher ranking than U19. Previous football-related research has indeed demonstrated relationships between match distance covered, VO_2MAX_, and league placement [58,59,60,61]. Higher running values might have therefore contributed to the U15′s superior placement. Other factors, such as squad and opponents’ playing level, as well as players’ technical proficiency will also affect tournament outcome [5,62,63]. Additionally, these differences might be exacerbated by the current competition context and study methodology, as well as inconsistent methodology and data reporting between studies [17].

The average RD was greater for these NZ cohorts (111 vs 109 m·min^−1^) when compared to a broader population sample, including senior and elite-level players [17]. Previous values found for male junior sevens players were also lower (103 ± 8 m·min^−1^) [20]. U16 and senior professionals in RU have been reported to cover just over half this youth sevens cohort’s RD [21,64]. In line with TD, U15 showed the highest running demands, in contrast to U19′s near-average RD, which is biased towards seniors [17]. Prior rugby research has documented differences in movement demands between school-age cohorts, while older youth demonstrate similar movement patterns to those in the senior game [21]. Most investigations conducted on male international sevens players indicate RD between ~79 and 112 m·min^−1^ for full matches, with values of 121 m·min^−1^ in select cases [17,44]. This variability demonstrates a strong context-dependency in sevens.

In line with general findings in the rugby codes, both age groups covered the majority of their distances at low speeds relative to their V_TOP_ [12,44,45,65,66,67,68]. On average, sevens youth movement speed concurs with that found in the sevens literature (6.5 km·h^−1^) [17]. This V_AVG_ surpasses that of national and international youth and senior RU performances, as well as some, but not all, elite-level sevens matches [17,21,49,69]. The U15 ran faster than U19 (~0.4 km·h^−1^) on average, and their average movement speed seems more variable. This is likely caused by the significant difference in sprints performed, which in turn might be related to differences in short-term and accumulated fatigue, due to substitution differences or prior in-season loading, which can impact neuromuscular performance [44,70]. Concurrently, conditioning and differences in power-to-weight ratio might play a role, as the U19 had considerably more body mass. However, V_MAX_ was similar for both sevens age groups and is comparable to that measured for U16 and U18 RU backs (27.0–28.4 km·h^−1^) [21]. These speeds approximate those of world-class sevens players’ average V_MAX_ during international matches (27.9 km·h^−1^) [45]. The similarities between age cohorts might indicate that V_MAX_ remains relatively unaffected in the academy pathway, despite more training years. Similar findings of small differences in V_MAX_ were reported by Clarke et al. (2016) for sevens males, across playing levels [20]. A higher body mass in older youth cohorts has been associated with lower sprinting velocity [71]. Considering the importance of sprinting and HIR for sevens performance, anthropometrics are an essential factor in the development of youth sevens players [63,72]. As previous research has demonstrated a relatively uniform performance standard in sevens players and between positions, a larger study sample is needed to determine if the academy pathway does indeed incrementally select for small variability of characteristics between-athletes [68,73].

Research suggests that higher amounts of HIR, VHSR, and sprinting can be indicative of performance outcome and playing level [63,74]. Kinematic categories have been found to be helpful in predicting success in sevens youth competition [19], but contradicting evidence shows that the relationship between kinematics and competition outcome is complex, and technical and tactical factors play a substantial role [57,75]. Approximately 16% of our cohort’s TD was covered at or above ~18 km·h^−1^, commonly referred to as HIR [17,29,76]. In spite of the teams’ tournament rankings in the current study, both groups performed similarly for HIRD, HSRD, and VHSRD. Only six studies on senior male sevens players found similar or greater HIRD (circa. > 18 km·h^−1^) and HSRD (circa. >20 km·h^−1^) [44,63,68,77,78,79]. VHSRD was at par with that of top eight SA elite junior squads (45.9 ± 45.1 m > 24.1 km·h^−1^) [19], and only accounted for 3% of TD.

The observed absolute sprint count in our investigation (19.4 ± 4.8) matches that of Premier Grade club RU players (12.6 ± 6.9–28.0 ± 8.6), despite considerably shorter playing bouts [80]. Moreover, the relative sprint count is eightfold that of adolescent RU match play (2.97 vs. 0.36 min^−1^) [81]. The higher sprint rate and consistent trivial and small differences seen in absolute HIR categories and HIacc/dec favor the more successful team (U15). This is in contrast to the work of Van den Berg et al. (2013), albeit possibly due to ambiguous reporting [19]. The HIacc, but not HIdec, count was higher than reported for international sevens matches (7.5 ± 2.0 vs. 3.9 ± 1.0 and 2.8 ± 1.1 vs. 3.1 ± 0.8 min^−1^) [45]. These values could indicate a difference in game dynamics between the youth and senior level. These loads should be taken into consideration, as maximal acceleration values can reach 3.6 ± 0.4 m·s^−2^ in youth [20]. The aforementioned HIR performances suggest high anaerobic taxation, and, in this case, might indicate suboptimal development of U19 glycolytic work capacity, compared to U15 [72].

Youth sevens players covered more distance jogging (z2) (~50% TD), compared to walking (z1) (~27% TD). This observation is counter to what Lee et al., 2022. established for senior sevens; in the course of four matches, significant between-match differences were seen in z2 (6.1–12 km·h^−1^), yet z2 distance (~25% TD) was consistently lower than z1 distance (0–6 km·h^−1^) (~34% TD) [16]. Of note here is the difference in methodology used (absolute vs relative thresholds), which impacts the ability to compare these measures in depth. When z2, z3 (12.1–14 km·h^−1^), and z4 (14.1–18 km·h^−1^) are combined, this hypothetical speed category (~49% TD) approximates that of the youth sevens. Indeed, a 6.1–18 km·h^−1^ speed band might approximate 20–50% V_TOP_, i.e., z2 used in the current study. This finding further highlights the importance of applying sport and population-specific, standardized movement categories [27,29,76]. In youth RU matches, the contrary results to sevens seems to be typical, as these players were shown to spend around three-quarters of game time stationary or walking, and jogged small amounts [25,81]. 

U15 sevens covered more distance *jogging* (~51% vs. ~50% TD) and *striding* or *cruising* (z3) (~22% vs. ~21% TD) than U19, as these movement categories are commonly labelled [65,66]. This difference in z2 and z3 running distance is possibly influenced by the discrepancy in games played (3 vs. 4); Lee et al., 2022. demonstrated a decrease in moderate-speed running bouts and increased collisions in match four, possibly leading to a lower TD [16]. Consequently, more than 70% of TD was performed at submaximal, yet considerable speeds, relative to players’ V_TOP_. In comparison, senior sevens covered ~15% of TD of their matches in z4 (14.1–18 km·h^−1^) and ~5% in z5 (18.1–20 km·h^−1^) [16]. In line with less successful youth sevens teams, U19 showed a trend towards walking (z1) more distance (~29% TD) versus U15 (~25% TD). These NZ youth players’ values differ from those found for SA U18 provincial sevens teams, who demonstrated more walking (35%), but less jogging (35%) and running (14%) [19]. 

The cardiovascular response during sevens games can be dyssynchronous from the kinematics due the game’s intermittent nature, slow HR kinetics, and various extraneous influences [57,82]. Very high cardiovascular taxation was nevertheless observed on average throughout these youth games. Even as compared to world class players (85.8 ± 4.95 %HR_MAX_), who saw significant blood metabolite accumulation measured by Couderc et al., 2017 [72]. Since substitute players tend to have higher movement variables, this could have influenced HR_AVG_ [44]. No differences were found between cohorts for cardiovascular load. This could indicate that HR might not accurately discriminate for match load in youth sevens age groups. More youth-specific sevens research is needed to clarify the role age plays in HR measures when compared to senior players. 

Cardiovascular taxation during these youth sevens matches does not seem to corroborate with U15′s “somewhat hard” perceived exertion (cf. 130 beats·min^−1^; cycle ergometer; 30–50 years) [24]. A similar observation was made by Blair et al. (2017) [57]. The Borg-scale holds useful aspects as a holistic indicator of psychophysical strain in simple and general circumstances [24]. Forms of RPE and HR monitoring can be used to monitor global internal exercise loads, e.g., in youth soccer matches [83,84]. Yet, RPE is context-dependent, and its relationship to HR in isolation is not consistently sufficiently strong; other factors, such as blood lactate levels strengthen its validity [24,83,85,86]. Additionally, it is reasonable to expect that fatigue will have affected performance to a small degree between first and second halves, but perhaps not significantly between matches [44,87]. Considering this one-day measurement however, the delayed onset of muscle damage and its implications on perceived exertion would not have manifested as a performance detriment [87]. Consequently, these factors might not be fully represented in post-match RPE. Compared to U15, elite-level sevens reported a higher sRPE^10^ (7) [57]. Moreover, it was argued that a tournament setting is suboptimal to collect RPE measurements, as multiple external factors might be of influence. Additionally, the possibility exists that younger players’ perceptions are insufficiently developed to adequately gauge physical load, and additional familiarization was needed. In this study, shorter playing bouts due to rolling substitutions could have affected players’ RPE [44]. No other youth sevens competition RPE data is available for further comparison. 

Our data demonstrates equal impact rates for both cohorts, yet lower than previously reported for junior, senior, and elite sevens players [20]. The upper range of recorded “body impacts” overlaps with average values reported for youth sevens [20]. This discrepancy could be due to a difference in evasion skills, and the lower impact rate might help explain the relatively low perceived exertion recorded, as RPE has been shown to significantly relate to the number of contacts in sevens [56]. However, differences in definitions, units, and thresholds used across studies lead to disparities observed in the literature, making comparisons difficult [29,37]. Since contacts are an inherent part of the game that should be differentiated between, accurate impact measures could be a meaningful and useful addition to training and match load [37,88]. By extension, AL_3D_ summarizes U15′s trend to undergo higher match loads than U19, which can be explained by this younger cohort’s higher values on several kinematic parameters. Proxy-measures of loading have been reported for RU youth (PlayerLoad rate 5.6 ± 0.9–7.3 ± 0.7) [21]. These indicators of external load can be a reference point within athletes or cohorts using the same devices. A critical perspective should be maintained when interpreting measures of impact and load [89]. In this regard, the limited impacts observed in our research are relative, and they are a product of the innovative VX SPORT™ approach, as an alternative to “G” measures, integrating threshold and discriminatory mechanisms to produce a meaningful impact perspective [55].

## 5. Conclusions

This novel study for the first time describes and differentiates the match demands of NZ male secondary school cohorts competing in national rugby sevens. A comparison is provided between two youth teams, as well as with their senior and international sevens peers and their RU peers. Within this rugby context, NZ youth sevens players seem to experience very high physiological and kinematic match loads, characterized by very high HR_AVG_ and continuous running demands, performed at moderate to high speeds. Younger players can cover more distance at a higher pace and perform more sprints in comparison to their older counterparts. Consequently, younger players can experience higher overall physiological, kinematic, and possibly, perceived workloads.

### 5.1. Practical Applications

Practitioners should be aware of the continuous high-intensity nature of youth sevens games; players mostly operate at or above jogging speeds, with sustained near-maximal HR. Training should be sevens-specific to reflect these competitive demands by improving both the aerobic and anaerobic capacity of the players, with particular attention paid to HIR and repeated sprinting. Differentiation between age groups is recommended, and an indication of perceived exertion should be trained within youth. Measures of internal and external load can be discrepant; coaches can use a combination of RPE, HR, and kinematic or physiologic measures to monitor competition to inform training.

### 5.2. Limitations

This study incorporated data from two school-age cohorts of modest sample size in seven matches during a national-level NZ-based sevens tournament. The interpretation of results must be done with reservation, considering the differences in monitoring equipment, as well as kinematic categories, and the disparity in contexts, methodologies, and data reporting in the literature [17]. Further youth-related sevens investigations are warranted to extrapolate and scale these initial conclusions.

## Figures and Tables

**Table 1 jfmk-08-00041-t001:** Age group sevens mean game movement demands.

	Pooled	U15		U19			
Variable	Mean ± SD	Mean ± SD	Range	Mean ± SD	Range	*p*	ES
Total distance (TD) (m)	1559.0 ± 166.2	1619.5 ± 186.6	[1078.0–1960.0]	1509.7 ± 147.7	[1232.0–1792.0]	<0.01 ^#^	Moderate
Relative distance (RD) (m·min^−1^)	111.4 ± 11.9	115.7 ± 13.3	[77.0–140.0]	107.8 ± 10.5	[88.0–128.0]	<0.01 ^#^	Moderate
Average speed (V_AVG_) (km·h^−1^)	6.7 ± 0.7	6.9 ± 0.8	[4.6–8.4]	6.5 ± 0.6	[5.3–7.7]	<0.01 ^#^	Moderate
Maximal speed (V_MAX_) (km·h^−1^)	27.6 ± 2.9	27.6 ± 2.7	[22.3–33.4]	27.7 ± 3.1	[19.2–33.7]	0.92	Trivial
HIRD (m)	251.7 ± 102.0	265.1 ± 99.7	[74.0–448.0]	240.8 ± 103.8	[20.0–514.5]	0.24	Small
HSRD (m)	165.7 ± 87.5	171.2 ± 84.0	[20.0–380.8]	161.3 ± 90.3	[0.0–430.5]	0.58	Trivial
VHSRD (m)	45.9 ± 51.3	48.5 ± 56.1	[0.0–204.0]	43.8 ± 47.0	[0.0–182.0]	0.66	Trivial
Sprints rate (per full game)	41.6 ± 7.6	44.3 ± 9.2	[28.0–63.0]	39.4 ± 6.1	[24.0–54.0]	<0.01 ^#^	Moderate
HIacc (per full game)	41.8 ± 11.6	42.3 ± 12.5	[12.0–62.0]	41.3 ± 10.7	[16.8–70.0]	0.68	Trivial
HIdec (per full game)	16.3 ± 6.4	17.3 ± 6.4	[4.0–31.5]	15.5 ± 6.5	[2.0–31.5]	0.17	Small
Average heart rate (%HR_MAX_)	90.0 ± 3.9	90 ± 3	[79.0–95.0]	89 ± 4	[70.0–95.0]	0.73	Trivial
ActivityLoad 3D rate (AL_3D_) (AU)	4.9 ± 1.0	5.2 ± 1.4	[0.0–7.1]	4.7 ± 0.6	[3.6–6.2]	0.06	Small
Impact rate (IR) (impacts·min^−1^)	0.07 ± 0.26	0.05 ± 0.21	[0.00–1.00]	0.09 ± 0.29	[0.00–1.00]	0.36	Trivial
RPE (6–20)	-	13 ± 1	[7.0–15.0]	-	-	-	-

^#^ Significant difference between groups; HIRD/HSRD/VHSRD = high-intensity, high-speed, and very high-speed running distance; HIacc/dec= high-intensity acceleration/deceleration; RPE= rating of perceived exertion.

**Table 2 jfmk-08-00041-t002:** Match distances covered in relative speed zones for sevens age groups (m).

	U15	U19		
V_Z_ (m)	Mean ± SD	Mean ± SD	*p*	ES
z1 (<20%V_TOP_)	402.1 ± 82.4 ^∀^	432.0 ± 77.5	0.07	Small
z2 (20–50% V_TOP_)	824.4 ± 165.8	750.5 ± 125.9	0.01 ^##^	Small
z3 (51–80% V_TOP_)	360.4 ± 122.6 ^∀^	315.3 ± 97.2	0.04 ^#^	Small
z4 (81–95% V_TOP_)	48.0 ± 47.0	41.1 ± 42.6	0.45	Trivial
z5 (>95% V_TOP_)	3.5 ± 7.4	5.7 ± 11.5	0.26	Small

^#^ Significant difference between groups; ^##^ Additional significance for Bonferroni-correction; ^∀^ No significant between zone difference within cohort for Bonferroni correction.

**Table 3 jfmk-08-00041-t003:** Time spent in relative heart rate zones for sevens age groups (% full game time).

	U15	U19		
HR_Z_ (%Full Game)	Mean ± SD	Mean ± SD	*p*	ES
z1 (<20%HR_MAX_)	0.3 ± 1.26	0.2 ± 1.0	0.93	Trivial
z2 (20–50% HR_MAX_)	1.4 ± 3.0	2.3 ± 6.8	0.44	Trivial
z3 (51–80% HR_MAX_)	8.0 ± 12.2	7.4 ± 11.1	0.70	Trivial
z4 (81–95% HR_MAX_)	32.3 ± 19.5	33.3 ± 22.1	0.81	Trivial
z5 (>95% HR_MAX_)	57.9 ± 24.1	56.8 ± 26.4	0.42	Trivial

## Data Availability

Anonymized data sets can be provided on reasonable request to corresponding author.

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
