# Peer review of "New Zealand Youth Rugby Sevens: A Comparative Match Demands Study"

_jfmk, 2023, doi:10.3390/jfmk8020041_

Round 1

Author Response

Dear reviewer,

Please find the our cover letter and corrections formulated to our original manuscript attached. The corrections have been applied, as stated, to the manuscript. But the document is still being optimised by addressing the other reviewers' requests. A definitive version will be uploaded when this is finalised.

We much appreciate your input. 

Kind regards,

Koen Wintershoven and colleagues.  

Reviewer 2 Report

Dear editor and authors,

Thank you for consider my profile for review the manuscript. I am glad to review a manuscript about youth and rugby training and particulary a non-studied topic. In general, I would like to congratulate the authors for their work and the quality of the manuscript. Nevertherless, i send you some coments in order to improve the paper:

Introduction

Good introduction.

Line 46- Physical contact could increase the athlete load perception. So, maybe take into account this paper could improve the sentence. Moreover, in the results (line 155) and in the discusión (line 273), you coment this issue.

Epp-Stobbe A, Tsai MC, Morris C, Klimstra M. The Influence of Physical Contact on Athlete Load in International Female Rugby Sevens. J Strength Cond Res. 2023 Feb 1;37(2):383-387. doi: 10.1519/JSC.0000000000004262. Epub 2022 Jul 1. PMID: 36696260.

Materials and methods (minor details):

Line 79: add space “range 14-15”

Line 80: add kg after “mass 88.0 ± 9.8”

Results:

Perfect.

Discussion:

Line 197 – the majority of he player covered their distance at low speed and U15 ran faster. Perhaps, you do not explain well why you have reported this data, because you only consider anthropometrics as an essential factor in de development of youth 7s. Maybe, one posible explanation could be the neuromuscular performance based on maximal strength and reactive strength. You can find more information on these references:

Poulos N, Haff GG, Nibali M, Graham-Smith P, Newton RU. Comparison of the Potentiating Effect of Variable Load Jump Squats on Acute Drop Jump Performance in Rugby Sevens Athletes. J Strength Cond Res. 2023 Jan 1;37(1):149-160. doi: 10.1519/JSC.0000000000004214. Epub 2022 Jan 20. PMID: 36515600.

Oliver JL, Lloyd RS, Whitney A. Monitoring of in-season neuromuscular and perceptual fatigue in youth rugby players. Eur J Sport Sci. 2015;15(6):514-22. doi: 10.1080/17461391.2015.1063700. PMID: 26366619.

Martin EA, Beckham GK. Isometric Mid-Thigh Pull Performance in Rugby Players: A Systematic Literature Review. J Funct Morphol Kinesiol. 2020 Dec 8;5(4):91. doi: 10.3390/jfmk5040091. PMID: 33467306; PMCID: PMC7804886.

Line 239 – Maybe a comparison between 7s could me more appropiate

Lee M, Soo J, Yeo V, Aziz AR, Ihsan M. Running demands and activity profile of men's rugby sevens: a tournament scenario. Biol Sport. 2022 Sep;39(3):529-535. doi: 10.5114/biolsport.2022.107023. Epub 2021 Jul 15. PMID: 35959342; PMCID: PMC9331338.

Line 266 – Maybe an explanation of the muscle damage and the movement patters could be reinforced.

Pereira LA, Nakamura FY, Moraes JE, Kitamura K, Ramos SP, Loturco I. Movement Patterns and Muscle Damage During Simulated Rugby Sevens Matches in National Team Players. J Strength Cond Res. 2018 Dec;32(12):3456-3465. doi: 10.1519/JSC.0000000000001866. PMID: 28240708.

Higham DG, Pyne DB, Anson JM, Eddy A. Movement patterns in rugby sevens: effects of tournament level, fatigue and substitute players. J Sci Med Sport. 2012 May;15(3):277-82. doi: 10.1016/j.jsams.2011.11.256. Epub 2011 Dec 19. PMID: 22188846.

Line 273 – as you mentioned in the manuscript, a difference in evasion skills could explain differences between groups. Could we hypothesize that a lower muscular strength and power performance could affect the results?

Redman KJ, Wade L, Whitley R, Connick MJ, Kelly VG, Beckman EM. The Relationship Between Match Tackle Outcomes and Muscular Strength and Power in Professional Rugby League. J Strength Cond Res. 2022 Oct 1;36(10):2853-2861. doi: 10.1519/JSC.0000000000003940. Epub 2021 Jan 15. PMID: 33470597.

Author Response

Dear Editor, dear reviewer

Thank you for your feedback and for receiving this cover letter with our accompanying manuscript corrections. Detailed below, you can find the revisions and/or replies, addressed in chronological order, to meet your concerns. These are as applied to the manuscript and in complement to those changes tracked in the revised version. The full manuscript is still under revision, to also meet the other reviewers’ remarks. As soon as the entire manuscript is revised, we will transfer the latest version for typesetting and follow-up review.

Many thanks for your efforts.

Kindest regards,

Koen Wintershoven

Reviewer 3 Report

Dear Authors,   The manuscript it is utilised microsensor technology but doesn't refer to the current knowledge about these technologies. The microsensor technologies that have been used appear outdated like the bibliography. Only a few references are from 2021 and none from 2022 or 2023. Once again appear evident that European and American references have been completely forgotten.  Please,  English language and style and fine spell check It is recommended.   Examples of updated bibliography  

McCormack S, Till K, Wenlock J, et al. Contributors to negative biopsychosocial health or performance outcomes in rugby players

(CoNBO): a systematic review and Delphi study protocol. BMJ Open Sport & Exercise Medicine 2022;8:e001440. doi:10.1136/ bmjsem-2022-001440

Epp-Stobbe A, Tsai MC, Morris C, Klimstra M. The Influence of Physical Contact on Athlete Load in International Female Rugby Sevens. J Strength Cond Res. 2023 Feb 1;37(2):383-387. doi: 10.1519/JSC.0000000000004262. Epub 2022 Jul 1. PMID: 36696260

Author Response

Dear Editor, dear reviewer

Thank you for your feedback and for receiving this cover letter with our accompanying manuscript corrections. Detailed below, you can find the revisions and/or replies, addressed in chronological order, to meet your concerns. These are as applied to the manuscript and in complement to those changes tracked in the revised version. The updated manuscript with all reviewers’ remarks addressed will be transferred to the editorial office immediately following the full revision.

Many thanks for your efforts.

Kindest regards,

Koen Wintershoven

Round 2

Reviewer 3 Report

Dear authors 

The manuscript has been drastically improved and has addressed all the points that were raised. I suggest a minor revision of the title. Please change "new zealand YOUTH RUGBY SEVENS: A COMPARATIVE 2 MATCH demands study" in New Zealand Youth Rugby Seven: A comparative 2 Match demands study.